# Lowering of the Neonatal Lung Ultrasonography Score after nCPAP Positioning in Neonates over 32 Weeks of Gestational Age with Neonatal Respiratory Distress

**DOI:** 10.3390/diagnostics12081909

**Published:** 2022-08-07

**Authors:** Alessandro Perri, Simona Fattore, Vito D’Andrea, Annamaria Sbordone, Maria Letizia Patti, Stefano Nobile, Chiara Tirone, Lucia Giordano, Milena Tana, Francesca Priolo, Francesca Serrao, Riccardo Riccardi, Giorgia Prontera, Giovanni Vento

**Affiliations:** 1Department of Woman and Child Health and Public Health, Fondazione Policlinico Universitario Agostino Gemelli IRCCS, 00168 Rome, Italy; 2Neonatal Intensive Care Unit, “San Giovanni Calibita Fatebenefratelli” Hospital, Isola Tiberina, 00186 Rome, Italy

**Keywords:** lung ultrasonography, respiratory distress syndrome, preterm infants

## Abstract

Respiratory distress (RD) is one of the most common causes of admission to the neonatal intensive care unit. Correct diagnosis and timely intervention are crucial. Lung ultrasonography (LU) is a useful diagnostic tool for the neonatologist in the diagnosis of RD; the neonatal lung ultrasonography score (nLUS) can be used in the diagnostic process, but some authors hypothesise that it is also useful for the management of some neonatal RD. The aim of this study is to analyse the changes in nLUS score before (T0) and after (T1) the start of respiratory support with nasal CPAP in neonates over 32 weeks of age with RD. Thirty-three newborns were enrolled in this retrospective study. LU was performed before and after the start of CPAP. The median nLUS scores at T0 and T1 were 9 (IQR 7–12) and 7 (IQR 4–10), respectively, and showed a significant difference (*p* < 0.001). The magnitude of reduction in nLUS score, expressed as a percentage, was inversely related to the need for subsequent administration of exogenous surfactant. The study suggests the usefulness of the nLUS score in assessing the response to CPAP in neonates over 32 weeks gestational age.

## 1. Introduction

Respiratory distress (RD) is one of the most common causes of neonatal intensive care unit (NICU) admission in preterm and term infants and is associated with acute and chronic adverse outcomes.

The underlying causes can vary; transient tachypnoea of the newborn (TTN), respiratory distress syndrome (RDS), pneumonia, meconium aspiration syndrome (MAS) and persistent pulmonary hypertension (PPHN) are among the most common [1].

Overall, RD occurs in up to 7% of newborns [2], more commonly in preterm infants, with incidence inversely proportional to gestational age [3,4,5]. 

In infants with a gestational age of over 32 weeks (GA), the differential diagnosis between TTN and RDS is difficult. TTN can be treated with oxygen therapy, RDS must be treated with respiratory assistance (RA), although non-invasive ventilation is preferable. Continuous positive airway pressure (CPAP) is mandatory, especially in the acute phase of RDS [6,7,8,9]. In addition, treatment of the severe form of this condition includes the administration of exogenous surfactant [10,11].

The need for ICU admission may complicate the management of these neonates, especially if transfer from a spoke centre to a hub centre is required [12]. Predicting the course of RD in neonates over 32 weeks GA is difficult. An objective diagnostic tool capable of detecting RD early in this particular group of neonates and monitoring treatment would be a useful tool for the neonatologist.

Lung ultrasonography (LU) in NICU is a safe, harmless, bedside diagnostic tool that has been shown to be more accurate than chest X-ray in diagnosing the major neonatal lung diseases. It is also a safe tool that does not use ionising radiation and can be repeated several times if necessary to provide timely information on the progression of the underlying pathology [13,14]. Its use has increased exponentially in recent years [15]. The standardisation of this examination, which by definition depends on the operator, has increased owing to the introduction of protocols and guidelines [16,17]. The main ultrasound signs of the different causes of neonatal respiratory disease are now widely described and allow accurate differentiation between them [18,19]. Lung ultrasonography has a high accuracy in the diagnosis of TTN [20]. The typical ultrasound image of TTN is represented by a symmetrical B-line distribution with a regular pleural line. The double lung point, the transition point between the normal part of the lung and the fluid-filled part, has high specificity for the diagnosis of TTN [21,22].

In contrast, neonates with RDS have an irregular and thickened pleural line, multiple hyperechoic subpleural consolidations, and widespread B-lines. In severe RDS, the lungs may appear completely white due to confluence of B-lines [23,24,25,26].

The neonatal lung ultrasonography score (nLUS) is a simple tool that has been shown to accurately predict the need for exogenous surfactant administration [27,28,29]. It assigns a severity score to pulmonary pathology based on ultrasound patterns of the anterior superior, anterior inferior, and lateral regions. For each area examined, a score of 0 is assigned if only A-lines are present; 1 if A-lines are present in the upper part of the lung and coalescent B-lines are present in the lower part of the lung or at least three B-lines are present; 2 if coalescent B-lines are present; 3 if extensive consolidations are present [28]. The approach does not include examination of the posterior lung areas, as the LUS score aims to screen critically ill patients with an examination that can be performed as quickly as possible and does not necessarily require mobilisation with the associated risk of destabilisation. 

The LUS score, performed in the first hour of life, also appears to be able to predict admission to the neonatal intensive care unit (NICU) for transient neonatal tachypnoea or respiratory distress syndrome in term and late-preterm infants [30].

To our knowledge, there are currently no studies comparing the nLUS score before and after the initiation of non-invasive respiratory support in patients with these characteristics. The aim of this study is to analyse the changes in nLUS score in neonates before initiation of respiratory support and during support with nasal CPAP (nCPAP) in order to improve neonatal care. The description of a possible pattern of worsening or improvement of the score may lead to early detection of neonates with a milder or more severe form of RDS and consequently to early treatment.

## 2. Materials and Methods

This is a retrospective study analysing a group of patients born at Fondazione Policlinico Universitario “A. Gemelli” IRCCS (Rome, Italy) and Fatebenefratelli Isola Tiberina-Ospedale San Giovanni Calibita (Rome, Italy) from December 2019 to January 2021. 

The primary aim of this study is to analyse changes in the nLUS score in neonates before the start of respiratory support and during support with nasal CPAP. The secondary aims are to analyse the relation between nLUS, oxygen requirement, clinical course of the underlying pulmonary pathology and to identify and analyse the difference in changes in nLUS score in infants intended to receive replacement therapy with exogenous surfactant versus those who will not need it.

We considered eligible for inclusion in the study every infant of gestational age over 32 weeks presenting respiratory distress at birth (Silverman score ≥ 3) and with nasal CPAP (nCPAP) needed in the first six hours of life, with an oxygen requirement greater than 25%. Exclusion criteria were: genetic or chromosomal abnormalities or major congenital malformations, congenital lung pathologies, congenital heart disease, absence of written informed consent to participate from parents/legal guardians, timing of ultrasound scans not respected. Within the first 3 h of life (T0), before the start of respiratory support with nCPAP, for any eligible infant, a lung ultrasonography was performed and the nLUS score was calculated. We repeated the exam at 4–6 h of life (T1), during respiratory support, according to our protocols. We reported and analysed the nLUS score, calculated at T0 and T1 (Figure 1). Any infants who needed respiratory assistance before the first scan was excluded from the study. For any eligible infants, written informed consent was obtained from parents or from legal guardians before enrolment. We collected data from clinical record about gestational age, LUS score at T0 and T1, difference between nLUS score at T0 and T1 (nLUS at T0–nLUS at T1) expressed as absolute number and percentage, level of nCPAP used, oxygen requirement, exogenous surfactant administration. We also collected and analysed data about sex, birth weight and antenatal steroids administration. 

Non-invasive respiratory assistance and the possible need to administer exogenous surfactant therapy were managed according to European Guidelines [31] and standardised internal protocols. The decision to start support with nCPAP was made in the presence of one of the following criteria: dyspnoea with Silvermann > 3, polypnea with respiratory rate > 75 breaths per minute, oxygen requirement of at least 30% to maintain saturation in the appropriate range, episodes of apnoea and bradycardia [32]. For all neonates, endotracheal surfactant was administered when oxygen requirements were greater than 30% despite optimisation of non-invasive respiratory care with nCPAP [33].

The nLUS has been previously validated in the neonatal field. Each lung was divided into three areas (upper anterior, lower anterior, lateral) and examined using a linear probe, frequency 12 MHz, through both transverse and longitudinal scans. Images were obtained using a LOGIQ E9 General Electrics ultrasound machine. For each lung area (upper anterior, lower anterior and lateral), a 0–3 score was given relating to lung’s echogenicity patterns Figure 2. The total nLUS (between 0 and 18) was obtained from the sum of the scores of the six areas studied. The execution of the ultrasound scans and the assignment of the relative LUS score were carried out by trained physicians. In order to minimise the discomfort of the infants during the examination, ultrasound scans were performed using pre-heated ultrasound gel. Non-pharmacological measures, such as non-nutritive sucking and gentle physical containment, were used to prevent patient agitation. All examinations were performed using sterile disposable probe covers, as established by internal protocols.

### Statistical Analysis

Data were analysed using IBM Statistical Package for Social Science 25.0 version (SPSS, Inc., Chicago, IL, USA). Normality of continuous data was evaluated using Shapiro–Wilk test. Because the data distribution was not normal, continuous data are expressed as median and interquartile range (IQR). The dichotomous variables are reported as absolute numbers and percentage. The nLUS score at T0 and T1 were compared using Wilcoxon test for related samples. We obtained the difference between nLUS score at T0 and T1 as absolute number and percentage. Oxygen requirement at T0 and T1 were also compared using Wilcoxon test for related samples. We tested correlation between CPAP level and improvement of nLUS score and between CPAP and oxygen requirement using Spearman correlation. Lastly, we divided the enrolled neonates into two groups, based on the need for therapy with exogenous surfactant, in order to compare the extent of the reduction in the nLUS score in the two groups. The numerical variation of the nLUS score was expressed as absolute number and as percentage and was compared using the Mann Whitney U test. A *p*-value < 0.05 was considered statistically significant.

## 3. Results

Thirty-three (33) neonates were included; median GA was 35 (IQR 34–37) and the median of their birth weight was 2530 g (IQR 2230–2993). About 20 neonates (60.6%) had a diagnosis of RDS, 6 neonates had TTN (18.2%), 7 neonates had pneumonia (21.2%). About 13 neonates (39.4%) received exogenous surfactant (Table 1), none of them needed a second dose of surfactant. A total of 13 (39.4%) were infants of diabetic mother.

Only 3 neonates (9.1%) showed worsening lung ultrasonography; 10 neonates (30.3%) remained stable and 20 (60.6%) showed improvement in lung ultrasonography, with significant changes in LUS score after the start of respiratory support with CPAP: the median of nLUS score at T0 was 9 (IQR 7–12), the median of nLUS score at T1 was 7 (IQR 4–10). Wilcoxon test showed a *p*-value < 0.001.

The median of the differences between nLUS at T0 and nLUS at T1 (nLUS T0-nLUS T1) was 2 (IQR 0–3). Levels of CPAP used were between 4 and 8 cm H_2_O, depending on the extent of respiratory distress and oxygen requirement. 

Median of oxygen requirement was 30% at T0 (IQR 25–30%) and 25% at T1 (22–30%). 19 neonates (58%) had a reduction in oxygen requirements at T1 compared to T0. The difference between oxygen requirement before and during respiratory assistance with CPAP was not significant: Wilcoxon test showed a *p* value of 0.48 (Table 2).

Spearman’s correlation showed an inverse relation between nLUS T0-nLUS T1 and oxygen requirement at T1, with a coefficient of −0.6 (*p* value 0.001). The extent of the reduction in the nLUS score expressed as a percentage was found to be inversely correlated with the need for subsequent administration of exogenous surfactant with a Spearman coefficient of −0.48 (*p* value 0.005). 

About 20 neonates had diagnosis of RDS and 13 received exogenous surfactant. Their gestational age was 34 (IQR 35–37), 6 (46%) were born from diabetic mothers and 8 (62%) were born from caesarean section. Their LUS score was 12 (IQR 7–12) at T0 and 10 (8–12) at T1. Neonates who needed surfactant therapy showed a change in the LUS from T0 and T1 equal to 0 (median; IQR 0–2); those who did not receive surfactant had a variation of 3 (median; IQR 0–6); Mann Whitney U test showed a significant difference between the groups (*p* 0.02) (Table 3). In Table 4 we reported the nLUS score related to underlying pathology.

## 4. Discussion

Our results showed an association between the short-term clinical improvement achieved by the use of early CPAP treatment for respiratory distress in neonates and the nLUS score. The majority of patients studied (60.6%) experienced a reduction in nLUS score and clinical improvement, as evidenced by a reduction in oxygen demand (58%) after initiation of respiratory support with CPAP. Current knowledge suggests that the improvement in oxygenation and respiratory mechanics can be explained physio-pathologically by greater alveolar distension with a consequent improvement in the ventilation-perfusion ratio and by the prevention of alveolar collapse. According to recent evidence, early CPAP treatment appears to have several advantages in the treatment of neonatal distress syndrome: By opening the lungs, residual functional capacity can be restored and maintained, reducing airway fatigue and preventing alveolar collapse and re-expansion (atelectotrauma). Early CPAP could also improve surfactant deficit [34]. The nLUS score appears to be effective in monitoring changes in the lungs after the start of respiratory support with CPAP, and we found that the score decreases significantly after the start of CPAP treatment. It also correlates with changes in the trend of oxygen demand.

The level of CPAP does not seem to correlate with the reduction in nLUS score: This seems to indicate a positive effect of non-invasive respiratory support independent of the level of pressure applied. However, with increasing pressure, oxygen demand is significantly reduced, showing a greater clinical benefit for higher pressure values in the range considered (4–8 cm H_2_O).

The nLUS score differed according to the underlying pathology: infants with RDS had higher nLUS score values than the others, especially in the group receiving therapy with exogenous surfactant. This confirms what has already been reported in the literature, namely that the nLUS score is able to predict the need for exogenous surfactant with good accuracy. Although the small sample size does not allow the development of a ROC curve and the consistent establishment of a cut-off, the infants receiving surfactant had higher LUS score values than the infants treated with non-invasive respiratory support.

The magnitude of the reduction in nLUS score, expressed as a percentage, was inversely correlated with the need for subsequent administration of exogenous surfactant.

We found an inverse relationship between nLUS T0-nLUS T1 and oxygen demand at T1. 

This is useful information to assist the clinician in making decisions about the management of the neonate with respiratory distress, especially when the decision involves transfer from a spoke centre to a hub centre, hospitalisation in the NICU and administration of exogenous surfactant. An objective diagnostic tool capable of early detection and monitoring of the progression of RD in this population of neonates would be a useful tool for the neonatologist and could reduce the time between the onset of the patient’s symptoms and their treatment.

The main limitations of the study are the retrospective study design, the small sample size and the heterogeneity of the characteristics (gestational age, neonatal weight, pathology, level of assistance needed) of the patients studied. In addition, a modified nLUS score has recently been proposed [35]. In the new score, the examination of the posterior lung fields is added. It would be interesting to repeat these data in a prospective study using the modified nLUS score. 

In summary, these are promising results suggesting the serial use of lung ultrasonography and the LUS score not only in the initial diagnosis but also in the monitoring of newborns with respiratory problems. Indeed, lung ultrasonography could be a valuable tool to assess in real time the actual improvement in lung status after starting respiratory support. It is an aid for the clinician to adjust management and subsequent support accordingly, increasing or decreasing as needed. In addition, lung ultrasonography can help the clinician to select patients who do or do not require administration of exogenous surfactant. In the last decade, some concerns have been raised about the harmfulness of ultrasound. However, these concerns should be taken with a grain of salt. There are only a few studies on this topic, mostly based on animal models. One of the most interesting is by Schneider-Kolsky et al. [36]. In this paper, pulsed Doppler ultrasound was found to be harmful to the brain of chicks and may impair cognitive functions. To our knowledge, these data have not been confirmed in any study on human models. In any case, it is right to limit the use of Doppler ultrasound on the brain to selected patients and diseases. No harmful effects have been found with ultrasound of the lungs. On the other hand, the ionising effect of X-rays on the chest is very well-known. At present, it is imperative to limit neonatal exposure to X-rays [37], and this goal can be achieved thanks to the increasing use of ultrasound and wireless ultrasound probes [38], which offer the highest level of safety in the diagnostic treatment of neonates, including those isolated for SARS-CoV-2.

## 5. Conclusions

The study suggests the usefulness of the nLUS score for assessing response to CPAP in neonates over 32 weeks GA. The nLUS score appears to decrease in infants who respond to CPAP; little or no decrease in nLUS score after CPAP may identify infants who require early administration of exogenous surfactant. Further studies are needed to confirm these findings.

## Figures and Tables

**Figure 1 diagnostics-12-01909-f001:**
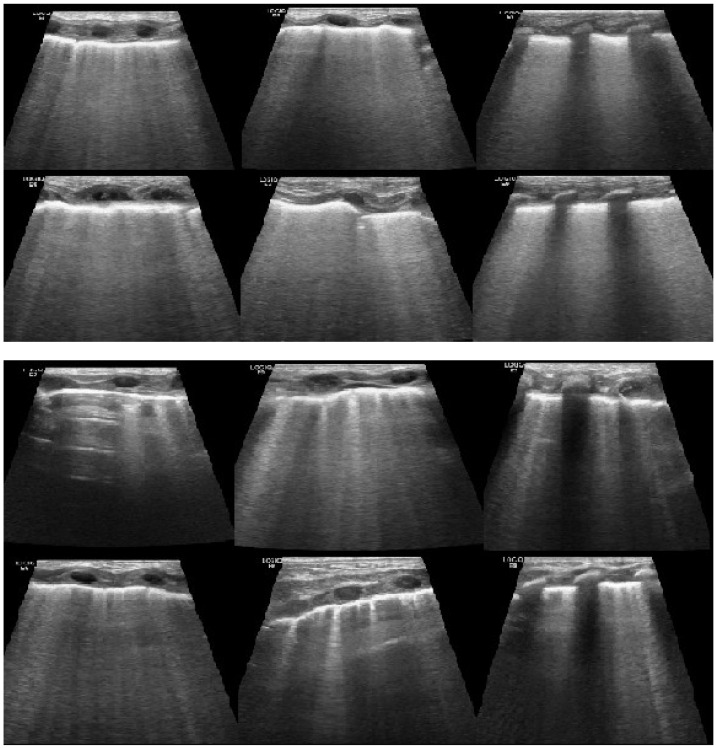
Sonograms at T0 and at T1 of a patient diagnosed with RD. After the positioning of the nCPAP the nlus score has lowered accordingly with the reduction of the coalescent b lines areas. In the second sonogram, the appearance of A lines can easily be noticed.

**Figure 2 diagnostics-12-01909-f002:**
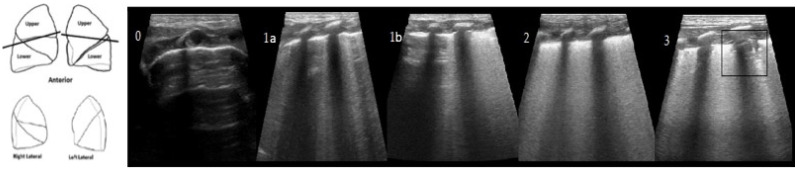
The lung ultrasonography score (nLUS). Lungs are divided into three areas: upper anterior; lower anterior; lateral. Each area is scored. Score values are related to the patterns that are shown in the upper part of the figure. Scores is given as follows: 0, only A-lines; 1a,b, presence at least 3 B-lines or A-lines in the upper part of the lung, coalescent B-lines in the lower part of the lung; 2, coalescent B lines with or without consolidations limited to sub-pleural space; 3, extended consolidation.

**Table 1 diagnostics-12-01909-t001:** Study population details.

	N = 33
GA (weeks)	35.4 (34.2–37.3)
Birth weight (grams)	2530 (2230–2993)
Vaginal delivery	11 (33.3%)
AGA	29 (87.9%)
SGA	2 (6.1%)
LGA	2 (6.1%)
Female	8 (24.2%)
Male	25 (75.8%)
Antenatal steroids	4 (12.1%)
No antenatal steroids	29 (87.9%)
RDS	20 (60.6%)
TTN	6 (18.2%)
Pneumonia	7 (21.2%)
Exogenous surfactant	13(39.4%)

Data are expressed as number (percentage) or median (IQR).

**Table 2 diagnostics-12-01909-t002:** nLUS, oxygen requirement before (T0) and after CPAP (T1).

	T0	T1	Z (Wilcoxon)	*p*-Value
nLUS	9 (7–12)	7 (4–10)	−3.66	<0.001
FiO_2_	30% (25–30%)	25% (22–30%)	−7	0.48

Data are expressed as median (IQR).

**Table 3 diagnostics-12-01909-t003:** nLUS reduction.

	Exogenous Surfactant (n = 13)	No Exogenous Surfactant (n = 20)	U (Mann Whitney)	*p*-Value
nLUSt0-nLUSt1	0 (0–2)	3 (0–6)	67.5	0.02
nLUSt0-nLUSt1(%)	0% (0–16%)	28%(4–77%)	58	0.007

Data are expressed as median (IQR).

**Table 4 diagnostics-12-01909-t004:** nLUS and diagnosis.

	LUS T0	LUS T1	DeltaLUS
RDS	10 (7–12)	8 (6–11)	1.5 (0–2.5)
TTN	8 (7–9)	2 (0–2)	5 (3.25–7)
Pneumonia	9 (7–10)	9 (7–10)	0 (0–0)

## Data Availability

The data presented in this study are available on request from the corresponding author.

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
