# Peer review of "Lowering of the Neonatal Lung Ultrasonography Score after nCPAP Positioning in Neonates over 32 Weeks of Gestational Age with Neonatal Respiratory Distress"

_diagnostics, 2022, doi:10.3390/diagnostics12081909_

Round 1

Reviewer 1 Report

Using unfamiliar abbreviation, such as nLUS, in the title without explaining what they stand for could make it hard for readers who are scanning through the titles. 

From the results, nLUS is not lowered in all neonates on  CPAP, as suggested in the title. Maybe a better way to re-phrase it would be more accurate.

Author Response

Using unfamiliar abbreviation, such as nLUS, in the title without explaining what they stand for could make it hard for readers who are scanning through the titles. 

From the results, nLUS is not lowered in all neonates on  CPAP, as suggested in the title. Maybe a better way to re-phrase it would be more accurate.

tankyou for your suggestion we change the title

Reviewer 2 Report

Dear Editor,

Thank you very much for giving me the chance to review the manuscript titled

“Lowering of the nLUS Score after nCPAP Positioning in Neonates over 32 Weeks of Gestational Age with Neonatal RDS” by Perri et al.

In this manuscript, the authors concluded that the nLUS score is useful to evaluate the response to CPAP in neonates over 32 weeks of gestational age.

I have major concerns regarding the manuscript:

-The sample size is very small in comparison to the time frame for the study and other published research in the same area.

-The study didn’t present new results to the written literature for example if compared to this study

 Brat R, Yousef N, Klifa R, Reynaud S, Shankar Aguilera S, De Luca D. Lung Ultrasonography Score to Evaluate Oxygenation and Surfactant Need in Neonates Treated With Continuous Positive Airway Pressure. JAMA Pediatr. 2015 Aug;169(8):e151797. doi: 10.1001/jamapediatrics.2015.1797. Epub 2015 Aug 3. PMID: 26237465.

Which was already published in 2015 and included both preterm and full-term neonates with different causes of respiratory distress

-In 2021 SzymaÅ„ski et al published a very interesting paper regarding modified nLUS score to include the posterior lung field and they concluded that in nonhomogeneous lung disorders like those presented in this research, the involvement of posterior lung fields seems even more significant. Therefore, posterior scans provide valuable information that should not be overlooked. Why the authors didn’t use this modified nLUS score?

zymański P, Kruczek P, Hożejowski R, Wais P. Modified lung ultrasound score predicts ventilation requirements in neonatal respiratory distress syndrome. BMC Pediatr. 2021 Jan 6;21(1):17. doi: 10.1186/s12887-020-02485-z. PMID: 33407270; PMCID: PMC7785923.

 Specific comments:

1) Title: This is misleading because the authors included in their work 40% of the studied population with other diagnoses rather than (RDS), which are TTN and neonatal pneumonia. Please change neonatal RDS to neonatal RD.

2) Abstract

The study suggests the usefulness of the nLUS score to evaluate the response to CPAP in neonates over 32 weeks of gestational age. Is that in patients with RDS or in other patients with other causes of RD, please specify?

3) Introduction:

-Is very long and needs to be summarized

- The approach does not provide the study of the posterior areas of the lungs because the intent for the LUS score is to study critically ill patients with an examination as fast as possible and which does not necessarily involve mobilization with the consequent risk of destabilization.  

In 2021 Szymański et al modified the nLUS score to include the posterior lung field and they concluded that in nonhomogeneous lung disorders like those presented in this research, the involvement of posterior lung fields seems even more significant. Therefore, posterior scans provide valuable information that should not be overlooked.

- As far as we know, there are no studies currently available that evaluate the LUS 83 trend after the start of CPAP

Please revise and use more conservative words.

4) Materials and methods:

-        The study was performed between December 2019 and January 2021, would you kindly mention how many deliveries per year in your hospital to include only 33 neonates in your study.

-        Did you include neonates with congenital heart diseases?

-        The first LUS performed after 3 or 6 hours of life?

-        Why did the authors select 4-6 hours of life to repeat the LUS score? Does this mean that sometimes they repeated the LUS after only 1 hour of using CPAP? Is that timeframe enough to assess the improvement or worsening of the case?

-        Why the authors didn’t use the most updated modified nLUS score to include the posterior lung in their study especially since they include a heterogeneous group of patients including 3 main diagnoses?

5) Results:

-        The median GA was 35 (IQR 34-37), why did the authors mention in the title of their work over 32 weeks and not over 34 weeks although all included neonates are 34 weeks or older?

-        Which are the 13% who received exogenous surfactant?

-        How many infants of diabetic mothers are included?

-        Only 3 neonates (9.1%) showed worsening in lung ultrasonography; 10 neonates  (30.3%) remained stable and 20 (60.6%) showed improvement in lung ultrasonography

Would you kindly describe in relation to the final diagnosis?

-        Would you kindly add a table regarding the nLUS scores in relation to the final diagnosis? Because the mentioned scores are high in comparison with the previously published data.

-        How many neonates need a second dose of exogenous surfactant? Did the authors perform follow-up LUS for these patients, if they do please mention that in the results part?

-        Would you kindly add LUS images for a patient who showed worsening in lung ultrasonography findings? And for a patient who showed improvement. This could help the reader to understand more about nLUS.

-        Would you kindly add the reliability data for LUS used to predict surfactant treatment in your patients

6) Discussion

-The discussion part is very short and needs to be re-written. The authors need to discuss their results in comparison with the already published data regarding:

·       The time frame they used for their follow up

·       The results for the nLUS score

·       The heterogeneity of the selected patients in this study

·       The relationship between LUS score and oxygen indices

·       The relation between LUS score and CPAP level

·       The reliability of the LUS scores to predict surfactant administration.

Reviewer 3 Report

This is an interesting study that asks an important question. The study is well designed and described. A tool that can predict infants who will need escalating respiratory support +/- surfactant +/- transport to another unit is useful.

Excluding the infants needing early CPAP likely excludes the more severe infants. Were the clinicians blinded to the score or did the ultrasound scores impact on the clinical treatment of the baby? eg Did an early high score result in infants being then quickly treated with CPAP +/- surfactant?

small suggestion - In the introduction it is stated that 'TTN can be treated with oxygen therapy'. In many centres infants with TTN will be treated with CPAP or high flow. 

Author Response

This is an interesting study that asks an important question. The study is well designed and described. A tool that can predict infants who will need escalating respiratory support +/- surfactant +/- transport to another unit is useful.

tankyou for your comment

Excluding the infants needing early CPAP likely excludes the more severe infants. Were the clinicians blinded to the score or did the ultrasound scores impact on the clinical treatment of the baby? eg Did an early high score result in infants being then quickly treated with CPAP +/- surfactant?

the clinicians were blinded to the nlus 

small suggestion - In the introduction it is stated that 'TTN can be treated with oxygen therapy'. In many centres infants with TTN will be treated with CPAP or high flow. 

Round 2

Reviewer 2 Report

Dear Editor,

I believe that the authors corrected all my suggestions and the manuscript is now ready for publication.

Author Response

Thank you for considering our article for publication